# *N*-Methyl-d-glucamine–Calix[4]resorcinarene Conjugates: Self-Assembly and Biological Properties

**DOI:** 10.3390/molecules24101939

**Published:** 2019-05-20

**Authors:** Ruslan R. Kashapov, Yuliya S. Razuvayeva, Albina Y. Ziganshina, Rezeda K. Mukhitova, Anastasiia S. Sapunova, Alexandra D. Voloshina, Victor V. Syakaev, Shamil K. Latypov, Irek R. Nizameev, Marsil K. Kadirov, Lucia Y. Zakharova

**Affiliations:** 1Arbuzov Institute of Organic and Physical Chemistry, FRC Kazan Scientific Center of RAS, 8 Arbuzov str., Kazan 420088, Russia; julianner@mail.ru (Y.S.R.); az@iopc.ru (A.Y.Z.); rezeda@iopc.ru (R.K.M.); anastasiya.strobykina@iopc.ru (A.S.S.); microbi@iopc.ru (A.D.V.); vsyakaev@iopc.ru (V.V.S.); lsk@iopc.ru (S.K.L.); irek.rash@gmail.com (I.R.N.); kamaka59@gmail.com (M.K.K.); lucia@iopc.ru (L.Y.Z.); 2Kazan National Research Technological University, 68 K. Marks str., Kazan 420015, Russia

**Keywords:** calixarenes, self-assembly, supramolecular chemistry, cytotoxicity

## Abstract

Deep insight of the toxicity of supramolecular systems based on macrocycles is of fundamental interest because of their importance in biomedical applications. What seems to be most interesting in this perspective is the development of the macrocyclic compounds with biocompatible fragments. Here, calix[4]resorcinarene derivatives containing *N*-methyl- d-glucamine moieties at the upper rim and different chemical groups at the lower rim were synthesized and investigated. These macrocycles showed a tendency to self-aggregate in aqueous solution, and their self-assembly abilities depend on the structure of the lower rim. The in vitro cytotoxic and antimicrobial activity of the calix[4]resorcinarenes revealed the relationship of biological properties with the ability to aggregate. Compared to macrocycles with methyl groups on the lower rim, calix[4]resorcinarenes with sulfonate groups appear to possess very similar antibacterial properties, but over six times less hemolytic activity. In some ways, this is the first example that reveals the dependence of the observed hemolytic and antibacterial activity on the lipophilicity of the calix[4]arene structure.

Academic Editor: J.A.A.W. Elemans

## 1. Introduction

Calixarenes are one of the important and promising classes of macrocyclic host-molecules in supramolecular chemistry. Thanks to their hard structure, presence of an inner cavity, wide possibilities of modification of the upper and lower rims, calixarenes have attracted considerable attention for various applications ranging from biochemistry to catalysis [1,2,3,4,5,6]. Both rims of these macrocycles can be modified for development of advanced functional molecules [7]. The addition of specific functional groups in the molecular structure influences the calixarene properties and, consequently, can provide new possibilities, among which are aggregation properties, aqueous solubility, biocompatibility, stimuli-response, biological activity, etc. [8,9,10]. Supramolecular aggregates of calixarenes formed by non-covalent interaction include micelles, vesicles, films and other nanostructures, which are used for different biological and industrial processes [1]. The non-toxicity of calixarenes allows the use of systems based on these macrocycles for delivery and to improve the bioavailability of drugs [3,11,12].

*N*-Methyl-d-glucamine (meglumine, MG) consists of tertiary amine fragments and sorbitol and has structural features related to glycosides. Its most interesting feature is the ability to form supramolecular adducts with lipophilic organic compounds in water [13]. This property is very useful in pharmaceutical industry for increasing the solubility of drugs and their stabilization in water solutions [13,14]. With meglumine derivatives as the amine component in Mannich reactions, potential pharmaceutical compounds have been attained [15]. The choice of *N*-methyl-d-glucamine as *N*-Mannich base is due to the important circumstance of increasing system stability. For example, substitution by *N*-methyl-d-glucamine increases the stability of cyclic peptide dimers and nanotubes [16]. The successful use of *N*-methyl-d-glucamine to obtain water-soluble macrocycles [17], namely porphyrin derivatives capable of localizing in cancer tissues [18] is also known. The high potential application of chlorin e6 noncovalently complexed with *N*-methyl-d-glucamine groups for photodynamic treatment was also shown [19].

In view of the above, it can be assumed that the introduction of the *N*-methyl-d-glucamine moiety into a macrocycle makes it possible to synergistically improve the benefits of both the calixarene platform and the glucamine fragment from the viewpoint of biomedical applications. Importantly, the *N*-methyl-d-glucamine group has never been used for modification of calix[4]resorcinarenes. Considering the biocompatible nature of meglumine group, we decided to modify the upper rim of calix[4]resorcinarene scaffold with this group and investigate the aggregation behavior of the macrocycles obtained depending on the lower rim structure. This paper presents data on the aggregation and biological properties of new calix[4]resorcinarenes, containing *N*-methyl-d-glucamine groups in the upper rim and different chemical groups (sulfonate and methyl) on the lower rim (Scheme 1). The revealed self-assembling and biological characteristics allow to form the basis for creation of nanoparticles with particular physicochemical properties suitable for medical applications.

## 2. Results and Discussion

### 2.1. Synthesis and Characterization of N-methyl-d-glucamine-Based Calix[4]resorcinarenes

Synthesis of *N*-methyl-d-glucamine-based calix[4]resorcinarenes (GCRs) was carried out in two steps as illustrated in Scheme 1, and the details are disclosed in the Materials and Methods section. At the first step, the synthesis of calix[4]resorcinarenes with sulfonate and methyl groups on the lower rim were carried out based on published methods [20,21]. Modification of their upper rim by *N*-methyl-d-glucamine fragments was performed by Mannich reaction in the *ortho*-position of resorcinarene cores, namely by three-component condensation of the calix[4]resorcinarene parent molecule, secondary amine (*N*-methyl-d-glucamine) and formaldehyde (Scheme 1). *N*-methyl-d-glucamine is mixed with paraformaldehyde and the mixture is heated until completely dissolution of the reagents, then a suspension of calix[4]resorcinarene in ethanol is slowly added while heating. In case of GCR-1 a water-methanol suspension is used. The reaction mixture was heated to 80 °C for 24 h. Raw products were filtered, purified by dialysis and recrystallized from ethanol. The structures of the synthesized compounds were confirmed by ^1^H-NMR, ^13^C-NMR and IR (Appendix A).

### 2.2. Conductometric Measurements

Firstly, self-assembly of GCR-1 was studied by conductometric measurements (Figure 1). Specific conductivity increases proportionately with the GCR-1 concentration in water solution. Inflection on the plot appears at 14 mM, and after this critical aggregation concentration (CAC) the increment rate of conductivity reduces because the aggregates transfer charge less efficiently than monomers. Remarkably, modification of calix[4]resorcinarene by *N*-methyl-d-glucamine fragment facilitates self-aggregation of macrocycle in water medium (Figure 1). The CAC value for calix[4]resorcinarene (CR-1) without meglumine units at the upper rim is 20 mM that is greater than that for meglumine-modified calix[4]resorcinarene. The driving forces of intermolecular aggregates’ formation are probably hydrogen bonds between hydroxyl groups on the GCR-1 upper rim, so the presence of additional OH-groups in the meglumine group reinforces this noncovalent interaction. A stable pH value, which can be observed over a wide range of concentrations, verifies that the *N*-methyl-d-glucamine fragment is an effective buffer component (Appendix A).

Since the hydrophobicity of GCR-2 is higher than that of GCR-1 due to presence of methyl groups instead of ionic sulfonate sites, water solutions of GCR-2 are unstable and eventually precipitate. As a result, the aggregation of this calix[4]resorcinarene was studied in water solutions at the presence of two-fold excess of *N*-methyl-d-glucamine (MG). Conductometic methods also confirm the self-assembly capability of GCR-2 in this aqueous medium. Conductometric values of CAC for GCR-2 of 7.9 mM are much lower than CAC of GCR-1, which is related with its lower water solubility due to the absence of ionic groups on the lower rim.

### 2.3. Investigation of Air-Water Interfacial Tension

In the next step of the study of GCR self-assembly the tensiometry method was used. Surface tension decreases with increasing GCR-1 concentration, but above the CAC (15 mM) the slope angle of the dependence changes (Figure 2). This reduction of liquid-air interfacial tension indicates that GCR-1 has a surface activity and propensity to aggregation. Concentration point of 15 mM, at which occurs inflection of surface tension dependence, corresponds to CAC that is in good agreement with conductometric value.

The tensiometric CAC value of GCR-2 (3.3 mM) is slightly different from the conductometric value (7.9 mM, Table 1). The tensiometry method is well known to fix the changes of the mutual disposition of the molecules at the water–air interface, whereas conductometry measures the number and mobility of ions in the bulk solution. The CAC values obtained for GCR-2 are lower compared to those for GCR-1. The improved ability to aggregation of GCR-2 in comparison with GCR-1 may be due higher hydrophobicity of GCR-2.

The presence of methyl groups on the lower rim close to the aromatic rims reduces the hydrophilic-lipophilic balance. It contributes to the formation of aggregates by means of Van-der-Waals interactions, in which intermolecular hydrogen bonds and π-stacking between aromatic rims of neighbor molecules of GCR-2 are possible.

### 2.4. UV-Spectroscopic Measurements

UV absorption spectrum of GCR-1 aqueous solution has a maximum absorption at 296 nm that corresponds to permissible π→π*-transition (Figure 3a). Also, the spectrum has a shoulder in the 500 nm region, which is indicative of the occurrence of zwitterionic structures in GCR-1 with transfer of protons from OH-groups to nitrogen. The same is observed in the case of GCR-2 (Figure 3a). 

Consequently, there is a chance that GCR-1 aggregates form not only by hydrogen bonds between hydroxylic groups but also by electrostatic interaction between negatively charged sulfonate groups on the lower rim and partially-positively charged meglumine groups on the upper rim. The change of optical density at 500 nm with increasing GCR-1 concentration has a nonlinear character (Figure 3b). The linear dependence breaks after 10 mM that correlates well with the CAC obtained by conductometry and tensiometry, and this violation of Lambert-Beer law is due to formation of GCR-1 aggregates in solution.

The method of solubilization of the hydrophobic dye Sudan I was used for determination of formed aggregates’ morphology. The appearance of an absorption band characteristic of a lipophilic substance in an aqueous solution of a specific component indicates the formation of a hydrophobic core of the component aggregates, like in surfactant micelles, which solubilizes the hydrophobic substrate. It was shown that increasing of GCR-1 concentration doesn’t lead to appearance of a Sudan I absorption band at the region of 500 nm (Figure 3b), i.e., solubilization of hydrophobic dye has not risen commensurately. It may testify that GCR-1 aggregates haven’t hydrophobic cavity, which could dissolve Sudan I. It also suggests that these aggregates have a stacked structure formed either by head-to-head pathway due to hydrogen bond between hydroxylic groups or by electrostatic head-to-tail interaction between negatively charged sulfonate groups of lower rim and positively charged upper rim.

Dependence of GCR-2 at 500 nm on its concentration also loses linearity at higher 7.2 mM (Figure 3b). This is because the dissolved molecules of GCR-2 aggregate that confirms by conductometry and tensiometry. The obtained values of the correlation coefficients r^2^ calculated for both calixarenes in a wide concentration range depict that GCR-2 has a greater ability to aggregate than GCR-1. GCR-2 solutions do not solubilize the hydrophobic Sudan I (Figure 3b), showing that formation of GCR-2 aggregates with hydrophobic cavity is non-identical with hydrophobic core of traditional surfactants. Probably, the short methyl groups fail to form a nonpolar core capable of incorporating hydrophobic guests. Thus, the cooperative effect of hydrogen bonds and π-stacking between GCR-2 molecules contribute to formation of compact aggregates incapable to solubilization of hydrophobic molecules. This is in line with our previous finding that difference between solubilization behavior of typical micelles and calix[4]resorcinol can be used as a tool for the control of binding/release behavior of lipophilic loads [22,23].

### 2.5. Self-Assembly Morphology

Formation of aggregates with increasing GCR-1 concentration was confirmed by the DLS method. In aqueous 1 mM GCR-1 solutions the formation of small particles with hydrodynamic diameter of 2.9 nm was observed (Figure 4a). This size is much more similar to that of a single molecule that to that of an aggregate. However, the bimodal distribution with peaks at 120 nm and 504 nm occurs for solution with 50 mM GCR-1. This could be attributed to the formation different stacked structures by head-to-head or head-to-tail interaction between GCR-1 molecules. For aqueous solutions of aminocalix[4]resorcinarene with sulfonate groups on the lower rim, the head-to-tail aggregation was also revealed previously as a result of the electrostatic interaction between oppositely charged rims [24]. Zeta-potential of GCR-1 aggregates obtained by electrophoretic light scattering in solution was −56.8 mV that correspond to sufficiently stable systems. Negative value was caused by dissociation of sulfonate groups of GCR-1 lower rim in aqueous medium. To confirm the morphology and dimension of the GCR-1 aggregates, TEM image was recorded (Figure 5a), where it was shown that GCR-1 molecules were assembled into non-well-defined large nanoparticles. The probable reason for such high polydispersity is multiple supramolecular interactions, such as Coulombic interactions between oppositely charged rims through head-to-tail self-assembly, hydrogen bonding between hydroxyl groups by head-to-head joining and π-π stacking between aromatic units of neighboring macrocycles (Figure 6a).

The DLS method showed that an increase of GCR-2 concentration in solution didn’t lead to significant increasing of particles’ sizes (Figure 4b). In a mixed 1 mM GCR-2–2 mM MG system particles with a hydrodynamic diameter of 2 nm were formed, which correlates well with the size of single macrocycle molecules. In a 15 mM GCR-2–30 mM MG mixed system particles with 3.8 nm diameter were formed. Such a size approximately corresponds to double the length of a GCR-2 molecule. This indicates of formation of spherical micelles by virtue of lateral packaging of aromatic walls by head-to-head contact due to π–stacking and hydrogen bonds between neighboring GCR-2 molecules as depicted in Figure 6b. Such an aggregation mode is similar to the self-assembly behavior described for other calixarenes [25]. The zeta-potential of obtained particles is −17.4 mV which is attributable primarily to hydroxylic groups. The formation of little spherical particles was further affirmed by TEM images (Figure 5b). In comparison with the nanostructure of GCR-1 aggregates, smaller nanostructures with a diameter of ca. 5 ± 2 nm were formed in the aqueous solution mixture of GCR-2, which was in good accordance with the DLS measurements.

### 2.6. Macroccyle Aggregation Comparison

The various physicochemical techniques were used to study the self-assembling properties of GCR-1 and GCR-2. However, due to different solubility of macrocycles, they were not investigated under the same conditions, since GCR-2 was studied in the presence of MG. In order to verify the difference in aggregation behavior of these macrocycles, we conducted additional experiments in 50% H_2_O–50% DMSO solution that ensured approximately the same limiting solubility of both GCR-1 and GCR-2. First, the conductometric dependencies on macrocycle concentration were obtained (Appendix A). However, for both calixarenes a linear increase in conductivity with an increase in the calixarene concentration was observed. Interestingly, in this case, the specific conductivity value for GCR-1 is above than that of GCR-2, which is probably related to the lack of MG molecules in GCR-2 solution. A similar pattern with a linear dependence is observed with a change in absorption at 500 nm for both macrocycles (Appendix A). The linear dependences of specific conductivity and optical density on the concentration of macrocycles probably indicate that the morphology of aggregates formed in the concentration range is unchanged. Thus, it is likely that the aqueous-organic medium contributes to the formation of aggregates at low concentrations of calixarenes.

NMR diffusion spectroscopy is a powerful tool to reveal aggregation behaviour of the supramolecular systems in solutions. Therefore, self-diffusion coefficients (D_s_) were measured for GCR-1 and GCR-2 at various concentrations (Figure 7). It turned out that for GCR-2 the D_s_ values are changed only slightly in a wide range of concentrations, however for GCR-1 the D_s_ value decreases with increasing concentration. In general, the D_s_ values for both macrocycles are similar, and the D_s_–C dependencies for them do not reflect a sharp phase transition revealed by conductometry and UV spectroscopy.

The relations between the surface tension and the concentration of macrocycle solution were illustrated in Figure 8. Although the surface tension values are lower than in the case of aqueous solutions, the form of γ–C dependences is approximately the same as in Figure 2. A decrease in the surface tension of GCR-1 is observed with an increase in its concentration with the formation of two plateaus. The first inflection point was determined after a concentration of 10 mM, which can be correlated with the CAC value in the aqueous environment. The second plateau is formed in a highly concentrated region with GCR-1 amount above 40 mM, which is likely due to either a change of aggregation modes or morphological reassembly of its aggregates. Hence, the different media can cause slightly different aggregation behavior of water-soluble GCR-1. The turning point of the tensiometric curve for GCR-2 can be determined as 3.3 mM that is same as in water. Thus, the tensiometry method turned out to be sensitive to changes in the concentration of GCR-2 in a mixed aqueous-organic medium and, in general, reproduced data obtained in an aqueous solution.

Since measurements of the surface tension of the macrocycles’ solutions revealed changes at the water–air interface, DLS was performed to investigate the effects of macrocycle concentration on the morphology of aggregates. Figure 9a showed the variation of size of GCR-1 with an increase of concentration from 5 mM to 50 mM. For 5 mM solution the hydrodynamic diameter is about 150 nm, which is increased to 190 nm with the GCR-1 concentration. The appearance of a bimodal distribution in concentrated solutions is caused by an increase in the index of polydispersity due to the formation of disordered structures. The change in the hydrodynamic diameter in GCR-2 solution is insignificant (from 3.4 nm to 4.2 nm) (Figure 9b), which correlates with DLS data obtained in an aqueous solution in the presence of MG.

Generally, the aggregation behavior of GCR-1 and GCR-2 is not exactly identical in both aqueous and aqueous-organic medium, and the inconsistency of the concentration dependence between aqueous and aqueous-organic solutions is due to several reasons. First, the different media cause different macrocycle aggregation behavior. Secondly, the measurements of self-assembly of ionic GCR-1 in organic media are much less informative compared to water due to the lower dissociation power of organic or water-organic media. Therefore, the number of ions in water will be different from the number of ions in an aqueous-organic medium, which will also be recorded differently by conductometry. Third, the contribution of cooperative interaction in the aqueous and aqueous-organic environment will also be different. Moreover, due to weaker solvophobic effect of water-organic mixtures compared to water dependences of the conductivity of solutions (as well as other properties) versus amphiphile concentration will be much less expressed, so that the breakpoints in the dependences can be very smooth if any. Thus, a detailed study of aggregation in a water-organic environment requires a separate careful investigation, which is beyond the scope of the presented work.

### 2.7. Evaluation of Toxicity and Biological Activity

The different types of antimicrobial activity tests of calix[4]resorcinarenes were conducted in vitro for this study. Table 2 contains data on bacteriostatic, fungistatic, bactericidal and fungicidal action against bacteria and fungi. The analysis of the data suggests that both calix[4]resorcinarenes selectively kill the Gram-positive bacterium *S. aureus* 209P. In the same time, GCR-2 has higher activity against *B. cereus* 8035. The antimicrobial activity of macrocycles appears at concentrations ranging from 0.13 to 1 mM, In general, calix[4]resorcinarenes were less toxic in relation to the cells of the test microorganisms, and their activity is due to the single molecule and not to the aggregates.

The evaluation of the cytotoxic effect of calix[4]resorcinarenes on human erythrocytes (hemolytic efficiency) has shown that these macrocycles exhibit toxic properties only at the high concentration (Table 3). At the low concentration GCRs show low hemolytic efficiency that correlates to the literature data [3,26]. Comparing both macrocycles under identical concentrations, it is worth noting that despite the ionic nature, sulfonated calix[4]resorcinarene is less toxic than a macrocycle containing a methyl group. Probably, such a difference in hemolytic activity is related to the aggregation ability of macrocycles. As also shown in Table 3, the compound concentrations inducing a 50% inhibition of cell growth (IC_50_ on Chang liver cell line) were more than 0.1 mM for each studied macrocycle. The liver cells were chosen as they have a well-established structure of rather uniform type. The cytotoxic effects as well as the antimicrobial properties were also related to the single molecule and not to the aggregates. Thus, the results indicate a low toxicity for macrocycles in low-concentration aqueous solutions and further recommend them for application elsewhere.

## 3. Materials and Methods

### 3.1. General Information

*N*-Methyl-d-glucaminemethylcalix[4]resorcinarenes were synthesized in two steps starting from calix[4]resorcinarenes with sulfonate and methyl groups on the lower rim obtained by a published method [20,21]. *N*-Methyl-d-glucamine (99%, Acros Organics, Fair Lawn, NJ, USA) and 1-phenylazo-2-naphthol (Sudan I, Acros Organics) were used as received. Sample solutions were prepared in the deionized water (18.2 MΩ) obtained from a Direct-Q 5 UV water purification system (Millipore, Molsheim, France). The accurate pH was measured with a HI 2110 pH meter (Hanna Instruments, Woonsocket, RI, USA) calibrated using buffers according to the manufacturer’s instructions.

### 3.2. Synthesis of N-methyl-d-glucaminemethyl Sulfonatoethylcalix[4]resorcinarene (GCR-1)

A mixture of *N*-methyl-d-glucamine (0.78 g, 4 mmole) and paraformaldehyde (0.12 g, 4 mmole) was stirred in 10 mL of methanol at 70 °C until the completely dissolution. Then, a solution of sulfonatoethylcalix[4]resorcinarene (1 g, 1 mmole) in water (10 mL) and methanol (50 mL) was added slowly in portions of 5 mL over 30 min. The reaction mixture temperature wasn’t allowed to rise above 65 °C. The precipitate formed after 24 h of heating was filtered, triturated in methanol and dialyzed (15 mL of an aqueous solution vs. 300 mL of water, 30 min × 3 times). Yield: 1.5 g (82 %). M.p. > 350 °C. ^1^H-NMR (D_2_O, 600 MHz): δ (ppm) = 7.27 (4H, H_ar._), 4.60 (4H, CH), 4.23 (16H, CH_2_N), 3.94–3.57 (24H, CHOH), 3.26 (12H, CH_3_N), 2.96 (8H, CH_2_SO_3_), 2.75 (8H, CH_2_CH_2_SO_3_). ^13^C-NMR (D_2_O, 150.9 MHz): δ (ppm) =1 56, 124, 122, 107, 70, 67, 62, 57, 53, 49, 39, 34. IR (KBr, cm^−1^): 3500–3000 (O-H), 1610, 1460 (C-C), 1180 (C-O), 1045 (S=O). Elemental analysis for C_68_H_104_N_4_Na_4_O_40_S_4_: found C 44.14, H 6.35, N 3.03, Na 5.11, S 6.49, calculated C 44.44, H 5.70, N 3.05, Na 5.00, S 6.98.

### 3.3. Synthesis of N-methyl-d-glucaminemethyl methylcalix[4]resorcinarene (GCR-2)

A mixture of *N*-methyl-d-glucamine (0.715 g, 3.7 mmole) and paraformaldehyde (0.110 g, 3.7 mmole) was stirred in ethanol (10 mL) at 70 °C until complete dissolution. Then, a solution of methylcalix[4]resorcinarene (0.5 g, 0.9 mmole) in ethanol (30 mL) was slowly added over 30 min. The reaction mixture was stirred at 70 °C for 24 h. The precipitate was filtered, dialyzed (15 mL of an aqueous solution vs. 300 mL of water, 30 min × 3 times) and recrystallized from ethanol. Yield: 1.1 g (88%). M.p. > 350 °C. ^1^H-NMR (D_2_O, 600 MHz): δ (ppm) = 7.38 (4H, H_ar._), 4.48 (4H, CH), 4.4–4.0 (16H, CH_2_N), 3.75–3.5 (24H, CHOH), 3.20 (12 H, CH_3_N), 1.71 (12H, CH_3_). ^13^C-NMR (D_2_O, 150.9MHz): δ (ppm) = 151, 125, 124, 102, 70, 68, 62, 57, 51, 29, 22 ppm. IR (KBr, cm^−1^): 3500–3000 (O-H), 1610, 1540 and 1460 (C-C), 1210 (C-O). Elemental analysis for C_64_H_100_N_4_O_28_: found C 55.89, H 7.50, N 4.09, calculated C 55.97, H 7.34, N 4.08.

### 3.4. Tensiometry

Surface tension measurements were conducted on a Du-Noüy tensiometer K6 equipped with platinum ring (KRÜSS, Hamburg, Germany). The tensiometer was calibrated against Milli-Q deionized water. The platinum ring was thoroughly cleaned and dried before each measurement. The measurements were done in such a way that the vertically hung ring was dipped into the liquid to measure its surface tension. It was then pulled out from the solution carefully. Each measurement was repeated until three consistent values (within ±0.5 mN∙m^−1^) were obtained.

### 3.5. Conductometry

Electrical conductivity measurements were carried out using an InoLab Cond 720 precision conductivity meter with a graphite electrode having a cell constant of 0.475 cm^−1^ ± 1.5%. Specific conductivity values were measured at least three times for each concentration. Values varying from each other by not more than 2% were taken into account. All samples were studied at 25 ± 0.1 °C.

### 3.6. Hydrophobic Dye Solubilization

Hydrophobic dye solubilization was carried out by adding an excess of Sudan I to solutions. These solutions were allowed to equilibrate for about 48 h at room temperature. They were filtered, and Sudan I absorbance was measured at 486 nm using a Specord 250 Plus spectrophotometer (Analytic Jena, Jena, Germany) using a 1 mm optical path length quartz cell. Dye absorbance was obtained by subtraction of the contribution of macrocycle to the summary spectrum. Each absorbance spectrum was obtained three times, and the absorbance intensities were within 2–3%.

### 3.7. Dynamic Light Scattering

The hydrodynamic diameters of the self-assemblies were obtained by dynamic light scattering on a Zetasizer Nano instrument (Malvern Instruments, Malvern, Worcestershire, UK). The source of the laser radiation was a He-Ne gas laser with a power of 4 mW and a wavelength of 632.8 nm. For zeta potential measurement Zeta potential Nano-ZS (Malvern) with laser Doppler velocimetry and phase analysis light scattering was used. The temperature of the scattering cell was controlled at 25 °C. Measurements were repeated at least five times. All scattering data were processed using Malvern Zetasizer Software 5.00 (version 5.00.).

### 3.8. Transmission Electron Microscopy

TEM images were recorded on a HT7700 TEM instrument (Hitachi, Tokyo, Japan) operated at 110 kV accelerating voltage. The samples with 20 mM solutions were ultrasonicated in water for 10 min and then dispersed on 300 mesh carbon-coated copper grid.

### 3.9. NMR Diffusion Spectroscopy

The Fourier transform pulsed-gradient spin-echo (FT-PGSE) experiments were performed by BPP-STE-LED (bipolar pulse pair–stimulated echo– longitudinal eddy current delay) sequence. Data were acquired with 150.0 ms diffusion delay, with bipolar gradient pulse duration from 3.0 to 4.0 ms (depending on the system under investigation), 1.1 ms spoil gradient pulse and a 5.0 ms eddy current delay. The bipolar pulse gradient strength was varied incrementally from 0.01 to 0.32 T/m in 16 steps. The diffusion experiments were performed at least three times and only the data with the correlation coefficients of a natural logarithm of the normalized signal attenuation (ln I/I_0_) as a function of the gradient amplitude b = γ^2^δ^2^g^2^(Δ-δ/3) (γ is the gyromagnetic ratio, g is the pulsed gradient strength, Δ is the time separation between the pulsed-gradients, δ is the duration of the pulse) higher than 0.999 were included. The temperature was set to 30 °C with a 535 l/h airflow rate in order to minimize convection effects. Experimental data were processed with the Bruker Xwinnmr software package (version 3.5). The diffusion constants were calculated by exponential fitting of the data belonging to individual columns of the pseudo 2D matrix. Single components have been assumed for the fitting routine. All separated peaks were analyzed and the average values were taken.

### 3.10. Antibacterial and Antifungal Activity

In vitro bacteriostatic and fungistatic activities of the macrocycles were evaluated against pathogenic representatives of Gram-positive bacteria (*Staphylococcus aureus* 209p, *Bacillus cereus* 8035), Gram-negative bacteria (*Pseudomonas aeruginosa* 9027, *Escherichia coli* F-50), and yeast (*Trichophyton mentagrophytes* var. gypseum 1773, *Aspergillus niger* BKMF-1119, *Candida albicans* 885–653). Minimal inhibitory concentrations (MICs) were estimated by conventional dilution methods for bacteria and fungi [27]. The antibacterial and antifungal assays were performed in Hottinger broth (HiMe-dia Laboratories Pvt. Ltd. Mumbai, India) and Sabouraud dextrose broth (HiMedia Laboratories Pvt. Ltd.) (bacteria 3 × 10^5^ cfu/mL and yeast 2 × 10^4^ cfu/mL). The bactericidal and fungicidal activities were determined as described earlier [28]. The tests were repeated three times.

### 3.11. Cell Viability Evaluation

Cell viability of human hepatocytes cells (Chang liver cell line from the N. F. Gamaleya Research Center of Epidemiology and Microbiology) toward macrocycles was determined by means of multifunctional system Cytell Cell Imaging (GE Healthcare, Issaquah, WA, USA) using application Cell Viability BioApp and Automated Imaging BioApp. The cells were dispersed on a 96-well plate at a concentration of 200,000 cells/mL and cultivated in CO2-incubator at 37 °C. Next, the culture medium was sampled in 24 h, and 150 μL of the studied dispersions was added to each well. The experiments were repeated three times. Intact cells cultivated simultaneously with the studied ones served as a reference. The fraction of the grown-up cells was expressed in % vs. reference cells. Degree of cell growth inhibition under the influence of testing agent was calculated by equation:N (%) = (1 − Exp/Control) × 100(1)
where Exp is the quantity of uninhibited cells in sample studied, Control is the quantity of uninhibited cells in control sample. Then the IC_50_ (concentration which caused 50% cell growth inhibition) was determined from curve of cell cultural growth versus macrocycle concentration. The experiments were repeated three times and results are presented as the mean ± standard deviation.

### 3.12. Hemolytic Activity

The hemolytic activity of the compounds was estimated by comparing the optical density of a solution containing a compound being tested and the blood with the optical density of the blood upon 100% hemolysis. A 10% suspension of human erythrocytes was used as an object of investigation; an erythrocytic mass with heparin was washed three times with a physiologic saline (0.9% NaCl) solution, centrifuged for 10 min at 800× *g*, and resuspended in the physiologic saline (0.9% NaCl) solution to a concentration of 10%. The concentrations of the compounds that corresponded to the MIC for the bacterial test strains were prepared in a physiologic saline (0.9% NaCl) solution (supplemented with 5% DMSO), and 4.5 mL of a compound at the corresponding dilution was added to 0.5 mL of a 10% suspension of erythrocytes. Samples were incubated for 1 h at 37 °C and centrifuged for 10 min at 2000× *g*. The release of hemoglobin was controlled by measuring the optical density of the supernatant on an AP-101 digital photoelectrocolorimeter (Apel, Kawaguchi, Japan) at 540 nm. Simultaneously, control samples were prepared: controls for zero hemolysis (blank): 0.5 mL of a 10% suspension of erythrocytes was added to a physiologic saline (0.9% NaCl) solution; 100% hemolysis: 0.5 mL of a 10% suspension of erythrocytes was added to 4.5 mL of distilled water. The experiments were repeated three times and data are presented as the mean ± standard deviation.

## 4. Conclusions

The calix[4]resorcinarenes were firstly derivatized by *N*-methyl-d-glucamine moieties at the upper rim to result in GCR-1 and GCR-2. The difference in structure of the lower rim of the studied macrocycles determines the different type of aggregation. The GCR-1 molecules form various large aggregates of the head-to-head and head-to-tail types due to the multicenter supramolecular interactions. The presence of methyl groups on the lower rim adjacent to the aromatic rings in the GCR-2 structure promotes the self-aggregation with formation of small spherical particles due to cooperative intermolecular hydrogen bonds between hydroxyl groups and π-stacking between the aromatic rings of adjacent molecules. This aggregation behavior of GCR-1 and GCR-2 is identical in both aqueous and aqueous-organic medium. Evaluation of hemolytic activity showed that the hemolytic effect of macrocycles decreases with decreasing concentration, and the degree of hemolysis for GCR-2 is greater than GCR-1. The antimicrobial activity of calix[4]resorcinarenes appears at concentrations from 0.13 to 1 mM. In general, these macrocycles are non-toxic, which will allow them to be used in biomedical applications.

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
