# Peer review of "N-Methyl-d-glucamine–Calix[4]resorcinarene Conjugates: Self-Assembly and Biological Properties"

_molecules, 2019, doi:10.3390/molecules24101939_

Round 1

Reviewer 1 Report

The work presented by Zakharova and collaborators describes the aggregation properties of two different calix[4]resorcinarene derivatives. Both the macrocycles are functionalized at the upper rim with four N-methyl-D-glucaminemethyl (MG) moieties, while the lower rim is functionalized with four sulfonate groups in case of GCR-1 and four methyl groups in case of GCR-2. The aggregation behavior of the two macrocycles in aqueous solutions or aqueous-organic mixture is studied at different concentrations. The last part of the manuscript describe in vitro antimicrobial activity tests.

In the introduction the Authors claim that the N-methyl-D-glucaminemethyl increases solubility of drugs and their stabilization in water solutions and that the non-covalent modification of of chlorin e6 with this meglumine group presents high potential application in photodynamic treatment. They decided to use MG to iduce aggregation in the resorcinarenes and to study the biological properties of these aggregates.

1. The biological properties of the aggregates are not clear. The Authors claim that the antimicrobial activity of the macrocycles appears at the concentration from 0.13 to 1mM, so below the calculated CAC values. For GCR-1 the DLS analysis confirmed that in water at 1 mM concentration the presence of small particles with hydrodynamic diameter of 2.9 nm was observed. This size is much more similar to the single molecule that to an aggregate. The antimicrobial activity presented by the Authors is due to the single molecule and not to the aggregates. At this point it is no more evident the advantage to have aggregates. The same for the cytotoxic effects. they were related to the single molecule, not to the aggregates, since the concentration of the compounds was too low to induce aggregartion.

2. The Author claimed that GCR-1 molecules were assembled into non-well-defined large nanoparticles, due to multiple supramolecular interactions, such as coulombic interactions between oppositely charged rims through head-to-tail self-assembly, hydrogen bonding between hydroxyl groups by head-to-head joining and π-π stacking between aromatic units of neighboring macrocycles. Such a conclusion must be supported by NOESY studies. NOESY NMR will help the Authors to clarify the structure of their aggregates. Same considerations are valid for GCR-2.

3. At page 4 lines 122-123 the Authors claim that GCR-1 presents a zwitterionic structure since a proton of one OH-group is transferred to the nitrogen. Why this is not observed also for GCR-2?

4. At page 5 lines 151-152 the Authors claim: “…size of inner aromatic cavity of GCR-2 is very small for formation 152 of host-gest complex with Sudan I ….”. This claim can be simply confirmed by computational calculations.

5. The two inflection points saw by the Authors during surface tension measurements for GCR-1 (page 7 and 8) can be ascribe to two different aggregations modes, head to tail and head to head. The Authors must comment on this.

6. At page 7 lines 227-229 the Authors claim: ”….the tensiometry method turned out to be sensitive to changes in the concentration of macrocycles in a mixed aqueous-organic medium and, in general, reproduced data obtained in an aqueous solution.” This is true only for GCR-2.

7. page 2 line 83: which is R in CR-1? Define it.

8. Typos:

page 2 line 69: N-methyl-D-glucamine is mixed with paraformaldehyde at the heating  and the mixture is heated until completely dissolving…

page 2 line 71: Reaction mixture was heating heated to 80°C for 24

page 7 line 210: self-diffusion coefficients (Ds) were measured for GCR-1 211 and GCR-2 at various concentrations (Figure 7).

General comment: all the figures must be readable also in black and white.

Author Response

The work presented by Zakharova and collaborators describes the aggregation properties of two different calix[4]resorcinarene derivatives. Both the macrocycles are functionalized at the upper rim with four N-methyl-D-glucaminemethyl (MG) moieties, while the lower rim is functionalized with four sulfonate groups in case of GCR-1 and four methyl groups in case of GCR-2. The aggregation behavior of the two macrocycles in aqueous solutions or aqueous-organic mixture is studied at different concentrations. The last part of the manuscript describe in vitro antimicrobial activity tests.

In the introduction the Authors claim that the N-methyl-D-glucaminemethyl increases solubility of drugs and their stabilization in water solutions and that the non-covalent modification of of chlorin e6 with this meglumine group presents high potential application in photodynamic treatment. They decided to use MG to iduce aggregation in the resorcinarenes and to study the biological properties of these aggregates.

1. The biological properties of the aggregates are not clear. The Authors claim that the antimicrobial activity of the macrocycles appears at the concentration from 0.13 to 1mM, so below the calculated CAC values. For GCR-1 the DLS analysis confirmed that in water at 1 mM concentration the presence of small particles with hydrodynamic diameter of 2.9 nm was observed. This size is much more similar to the single molecule that to an aggregate. The antimicrobial activity presented by the Authors is due to the single molecule and not to the aggregates. At this point it is no more evident the advantage to have aggregates. The same for the cytotoxic effects. they were related to the single molecule, not to the aggregates, since the concentration of the compounds was too low to induce aggregartion.

Answer: Yes, you are right. Relevant comments have been inserted into discussions about antimicrobial and cytotoxic properties in the revised manuscript.

2. The Author claimed that GCR-1 molecules were assembled into non-well-defined large nanoparticles, due to multiple supramolecular interactions, such as coulombic interactions between oppositely charged rims through head-to-tail self-assembly, hydrogen bonding between hydroxyl groups by head-to-head joining and π-π stacking between aromatic units of neighboring macrocycles. Such a conclusion must be supported by NOESY studies. NOESY NMR will help the Authors to clarify the structure of their aggregates. Same considerations are valid for GCR-2.

Answer: We obtained NOESY spectrum for a similar resorcinarene with sulfo groups on the lower rim and another amino group on the upper rim (J Phys. Chem. C, 2013, 117 (39), pp 20280–20288). With a concentration of this macrocycle equal to 80 mM, we can see the cross-peaks between adjacent protons inside a separate molecule, but not between the molecules. Therefore, this method is probably more suitable for confirming the cone-type (rccc) conformation of macrocycle molecule, and not aggregate structure.

3. At page 4 lines 122-123 the Authors claim that GCR-1 presents a zwitterionic structure since a proton of one OH-group is transferred to the nitrogen. Why this is not observed also for GCR-2?

Answer: Figure 3A contains the UV spectrum for GCR-2, where a shoulder in the 500 nm region is also observed. We introduced this comment into the text.  

4. At page 5 lines 151-152 the Authors claim: “…size of inner aromatic cavity of GCR-2 is very small for formation 152 of host-gest complex withSudanI ….”. This claim can be simply confirmed by computational calculations.

Answer: Thank you for your comment. We re-write this phrase. Our experience with surfactants has shown that Sudan I can be encapsulated with micelles formed from surfactants with long alkyl tails. This difference between solubilization behavior of typical micelles and calix[4]resorcinol was used as a tool for the control of binding/release behavior of lipophilic loads [PCCP 2011, J Phys. Chem. C , 2013, 117 (39), pp 20280–20288]. Importantly, the solubilization of the probe in the aquatic environment multiply increases with its encapsulation after CMC. In our work, even if there is a slight increase in theSudanI absorption, this change is within the limits of error.

5. The two inflection points saw by the Authors during surface tension measurements for GCR-1 (page 7 and 8) can be ascribe to two different aggregations modes, head to tail and head to head. The Authors must comment on this.

Answer: It is difficult to determine the aggregation mode by tensiometric dependencies, reflecting the change at the water-air interface. Both aggregation modes can only be determined by NMR, but in our case it is impossible. Although we agree with you that presence of inflection points on tensiometric dependence may be associated with a change in the aggregation mode and inserted this point in the revised manuscript.

6. At page 7 lines 227-229 the Authors claim: ”….the tensiometry method turned out to be sensitive to changes in the concentration of macrocycles in a mixed aqueous-organic medium and, in general, reproduced data obtained in an aqueous solution.” This is true only for GCR-2.

Answer: Thank you for your comment. We corrected this discussion in the revised manuscript.

7. page 2 line 83: which is R in CR-1? Define it.

Answer: This abbreviation is clarified in the revised manuscript.

8. Typos:

page 2 line 69: N-methyl-D-glucamine is mixed with paraformaldehyde at the heating  and the mixture is heated until completely dissolving…

page 2 line 71: Reaction mixture was heating heated to 80°C for 24

page 7 line 210: self-diffusion coefficients (Ds) were measured for GCR-1 211 and GCR-2 at various concentrations (Figure 7).

Answer: Thanks for the comments. All errors are eliminated in the revised manuscript.

General comment: all the figures must be readable also in black and white.

Answer: We corrected Figs 2,3,7 and 8 in the revised manuscript.

Reviewer 2 Report

This manuscript needs to be strengthened as follows:

1.       In the introduction part, I recommend the authors clearly state the problems that this study was going to address and the significance of performing this study.

2.       In the Results and Discussion part, the authors described the results and proposed the possible reasons to explain the results. However, the proposed explanation lack literature support. The authors should cite relevant references to support the discussion.

3.       In figure 5, I recommend the authors use arrows to point out the aggregates in those two images.

4.       The authors need to clarify why the Chang cells were chosen for the cytotoxicity study.

Author Response

This manuscript needs to be strengthened as follows:

1.       In the introduction part, I recommend the authors clearly state the problems that this study was going to address and the significance of performing this study.

Answer: The problem was added to the Introduction as: “The N-methyl-D-glucamine group is promising for the modification of functional substances, however it has never been used for modification of calix[4]resorcinarenes.”

The significance of performing this study was already indicated at the end of the introduction as: “The revealed self-assembling and biological characteristics allow to form the basis for creation of nanoparticles with particular physicochemical properties suitable for medical applications.”

2.       In the Results and Discussion part, the authors described the results and proposed the possible reasons to explain the results. However, the proposed explanation lack literature support. The authors should cite relevant references to support the discussion.

Answer: The work is devoted to the study of the aggregation behavior and biological characteristics of new calixarenes. We assume the mechanisms of aggregation based on the data obtained by physicochemical methods and correlate them with the literature data, in particular, reference 19 for GCR-1 and reference 21 for GCR-2.

3.       In figure 5, I recommend the authors use arrows to point out the aggregates in those two images.

Answer: Arrows is now added to the figure 5 in the revised manuscript.

4.       The authors need to clarify why the Chang cells were chosen for the cytotoxicity study.

Answer: The Chang liver cells were chosen as they have a well-established structure of rather uniform type. This point has been inserted in the revised manuscript.

Reviewer 3 Report

The paper submitted by Kashapov and co-worker is complete characterization of two different N-methyl-D-glucamine–Calix[4]resorcinarene. The manuscript is well organized and the large amount of experiments is consistent with the conclusions suggested by the authors. However some points have to be clarified before accepting the manuscript for publication:

1)  References must be improved: in the introduction between references 1 and 6 the authors should be add at least Chem. Soc. Rev. 2013, 42, 366 andSupramolecular Chemistry 2016, DOI: 10.1080/10610278.2015.1125900. Then a nice review should be add for resorcin[4]arenes: RSC Adv., 2015, 5, 51919.

2)  The final structure of calix GCR-1 and 2 should be chiral indeed the carbon with the first OH- group is a stereogenic centre[ -NHCH2-C(H)(OH)-CHOH-]. Did the authors consider that point? Did the authors separate the all diastereoisomers formed? If the authors have separated the enantiomers they should report the CD spectra

3)  At page 4 line 122 the authors ascribe the absorption band at 500 nm to the zwitterionic structure of GCR-1. There are some experimental data confirming this sentence? They could report Uv spectra at different pH in the supplementary materials, whose should show different intensity for the band at 500 nm.

4)  Which is the concentration used for TEM experiments. In general are missed important information in the materials and methods session on the samples used for each techniques.

5)  Why the authors did perform the DLS experiments of the GCR-2 in the presence of MG? In my opinion the logical message is missed

6)  At page 8 line 241 the authors conclude that the aggregation behaviour of GCR-1 and GCR-2 is identical in both aqueous and aqueous-organic medium; however comparing figure 3b and figure S6 seem that the aggregation behaviour is completely different, could the authors explain what is wrong?

Author Response

The paper submitted by Kashapov and co-worker is complete characterization of two different N-methyl-D-glucamine–Calix[4]resorcinarene. The manuscript is well organized and the large amount of experiments is consistent with the conclusions suggested by the authors. However some points have to be clarified before accepting the manuscript for publication:

1)  References must be improved: in the introduction between references 1 and 6 the authors should be add at least Chem. Soc. Rev. 2013, 42, 366 andSupramolecular Chemistry 2016, DOI: 10.1080/10610278.2015.1125900. Then a nice review should be add for resorcin[4]arenes: RSC Adv., 2015, 5, 51919.

Answer: References have been inserted in the revised manuscript.

2)  The final structure of calix GCR-1 and 2 should be chiral indeed the carbon with the first OH- group is a stereogenic centre[ -NHCH2-C(H)(OH)-CHOH-]. Did the authors consider that point? Did the authors separate the all diastereoisomers formed? If the authors have separated the enantiomers they should report the CD spectra

Answer: We used N-methyl-D-glucamine in the synthesis of calixarenes, and the same configuration of N-methyl-D-glucamine is preserved on the upper rim of the calixarene cup

3)  At page 4 line 122 the authors ascribe the absorption band at 500 nm to the zwitterionic structure of GCR-1. There are some experimental data confirming this sentence? They could report Uv spectra at different pH in the supplementary materials, whose should show different intensity for the band at 500 nm.

Answer: Previous experience with other aminomethylated calix[4]resorcinols with sulphonate groups on the lower rim showed the presence of a zwitterionic structure at the natural pH of the medium (New J. Chem. 2009, 33, 2397–2401; J Phys. Chem. C , 2013, 117 (39), pp 20280–20288). The same patterns of changes in the UV spectra at different pH are always reproduced for other aminomethylated calix[4]resorcinols.

4)  Which is the concentration used for TEM experiments. In general are missed important information in the materials and methods session on the samples used for each techniques.

Answer: This information is now added in the revised manuscript. Low concentration (0.5 mM) solutions were dispersed on 300 mesh carbon-coated copper grid due to the high ability of calixarenes to form a film, which greatly degrades the quality of the image.

5)  Why the authors did perform the DLS experiments of the GCR-2 in the presence of MG? In my opinion the logical message is missed

Answer: We hope that the macrocycles synthesized have the potential to be used in in vivo experiments, so it was important to investigate their aggregation in the aqueous environment, since water-soluble form of GCR-2 is obtained in the presence of MG and investigated by DLS.

6)  At page 8 line 241 the authors conclude that the aggregation behaviour of GCR-1 and GCR-2 is identical in both aqueous and aqueous-organic medium; however comparing figure 3b and figure S6 seem that the aggregation behaviour is completely different, could the authors explain what is wrong?

Answer: We corrected this point as follows: Generally, the aggregation behavior of GCR-1 and GCR-2 is not exactly identical in both aqueous and aqueous-organic medium, and the inconsistency of concentration dependences between aqueous and aqueous-organic solutions is due to several reasons. First, the different media cause different aggregation behavior of macrocycles. Secondly, the measurements of self-assembly of ionic GCR-1 in organic media are much less informative compared to water due to the lower dissociation power of organic or water-organic media. Therefore, the number of ions in water will be different from the number of ions in an aqueous-organic medium, which will also be recorded differently by conductometry. Third, the contribution of cooperative interaction in the aqueous and aqueous-organic environment will also be different. Moreover, due to weaker solvophobic effect of water-organic mixtures compared to water dependences of the conductivity of solutions (as well as other properties) versus amphiphile concentration will be much less expressed, so that the breakpoints in the dependences can be very smooth if any. Thus, a detailed study of aggregation in a water-organic environment requires a separate careful investigation, which is beyond the scope of the presented work.

Round 2

Reviewer 1 Report

The manuscrit resulted now improved

Reviewer 3 Report

The authors have replayed in the correct way to all points and provided a new version of the manuscript, which now is suitable for publication.